# Oribron: An Origami-Inspired Deformable Rigid Bronchoscope for Radial Support

**DOI:** 10.3390/mi14040822

**Published:** 2023-04-06

**Authors:** Junjie Su, Yangyang Zhang, Liang Cheng, Ling Zhu, Runhuai Yang, Fuzhou Niu, Ke Yang, Yuping Duan

**Affiliations:** 1School of Biomedical Engineering, Anhui Medical University, Hefei 230009, China; sujunjiesdut@163.com (J.S.);; 2Anhui Institute of Optics and Fine Mechanics, Hefei Institutes of Physical Science, Chinese Academy of Sciences, Hefei 230031, China; 3School of Mechanical Engineering, Suzhou University of Science and Technology, Suzhou 215009, China

**Keywords:** waterbomb origami, rigid bronchoscope, radial support, pneumatic control

## Abstract

The structure of a traditional rigid bronchoscope includes proximal, distal, and body, representing an important means to treat hypoxic diseases. However, the body structure is too simple, resulting in the utilization rate of oxygen being usually low. In this work, we reported a deformable rigid bronchoscope (named Oribron) by adding a Waterbomb origami structure to the body. The Waterbomb’s backbone is made of films, and the pneumatic actuators are placed inside it to achieve rapid deformation at low pressure. Experiments showed that Waterbomb has a unique deformation mechanism, which can transform from a small-diameter configuration (#1) to a large-diameter configuration (#2), showing excellent radial support capability. When Oribron entered or left the trachea, the Waterbomb remained in #1. When Oribron is working, the Waterbomb transforms from #1 to #2. Since #2 reduces the gap between the bronchoscope and the tracheal wall, it effectively slows down the rate of oxygen loss, thus promoting the absorption of oxygen by the patient. Therefore, we believe that this work will provide a new strategy for the integrated development of origami and medical devices.

## 1. Introduction

A rigid bronchoscope (RBs) [1] is an interventional surgical instrument that can treat respiratory diseases, such as tracheal obstruction [2]. The traditional RBs consist of proximal, distal, and body, as shown in Figure 1a: (1) the proximal contains a number of open ports for the simultaneous connection of external devices, such as ventilators and light systems. (2) the distal is a slope to facilitate RBs through a narrow area. There are side holes near the distal to maintain oxygen supply to the lungs, while RBs are inserted into the trachea. (3) the body is between the proximal and distal; it is made of hollow tubes of equal diameter. However, part of the oxygen will inevitably escape from the side hole when supplying oxygen, resulting in a low utilization rate of oxygen, as shown by (i) in Figure 1c. This is undoubtedly fatal for patients, as prolonged hypoxia can lead to brain damage and even death [3]. In the current study, there are mainly the following methods that can improve the utilization rate of oxygen: high-flow oxygen therapy, oxygen humidifier, oxygen mixer, oxygen delivery device, and pulmonary expansion agent. These methods can improve the utilization rate of oxygen, and control the flow rate of oxygen better. However, there is often the need to pay for additional treatment to install support devices and they often lack compatibility. A comparison of these methods can be found in the Appendix A (see Appendix A).

To solve this problem, this work proposed deformable RBs, which can improve blood oxygen mainly by reducing oxygen loss, as shown in Figure 1b. The deformable structure has the advantages of flexibility, modularity, and extensibility [4,5], and can undergo geometric changes in specific directions under the stimulation of gas [6,7,8,9,10], optical [11], thermal [12], magnetic [13], and electrical conditions [14], so it is suitable for a variety of complex unstructured environments [15,16]. However, at present, the fabrication of deformable structures is usually operated on a three-dimensional scale, which has the disadvantages of low precision, long duration, and high cost [17,18,19]. 

Origami is an ancient technique that converts two-dimensional crease patterns (CPs) into three-dimensional deformable structures by folding and deploying [20], it can rapidly change the size, shape, and motion of objects [21,22], which has the advantages of reconfigurability [23], variable stiffness [24], multi-stability [25] and negative Poisson [26]. Therefore, origami has been widely used in surgical robots [27,28], soft robots [29,30], microrobots [14,31], and mechanical metamaterial [32,33] in recent years. By changing the arrangement of CPs, scientists have designed many classical origami CPs, including Waterbomb [34,35,36], Kresling [37,38,39], Yoshimura [25,40], Miura [41,42], Flasher [43,44], and Square-twist [45,46]. Among them, Waterbomb has two different motion paths [47], so it has great attention. One is positive Poisson deformation with an antagonistic relationship in axial and radial directions, and the other is negative Poisson with simultaneous expansion/contraction. Therefore, Waterbomb is ideal for situations where radial variations are coupled to the surroundings, such as variable wheel [48,49], worm-like robot [50,51], and medical stent [52,53]. Inspired by Waterbomb’s positive Poisson mechanism, we want to attach Waterbomb to the body of RBs to form Oribron, as shown in Figure 1a. *P*_0_ is defined as the initial pressure when the proximal enters oxygen. When Oribron enters or exits the trachea,
*P*_0_ = 0(1)
the Waterbomb is a compact configuration with a small diameter (small form), as shown by (ii) in Figure 1c. When Oribron is working,
*P*_0_ > 0(2)
the Waterbomb is an expansion configuration with a large diameter (large form), as shown by (iii) in Figure 1c. At this point, the Waterbomb has infinitely close to the trachea to almost completely block the escape of oxygen.

In this study, we report deformable RBs with radial support capability. The deformable structure is made by the origami method, which has the advantages of lightweight, simple manufacture, and low cost, and has a large radial deformation ratio under pneumatic control. We evaluate the feasibility of combining origami with RBs in five ways: (1) geometric design, (2) kinematic modeling, (3) mechanical analysis, (4) pneumatic control, and (5) a smoke experiment.

## 2. Materials and Methods

### 2.1. Fabrication of the Waterbomb

In this study, we chose some lighter thin films (thickness ≤ 50 μm) as origami material, including PET (Jubang Plastic Material Co., Ltd., Dongguan, Guangdong, China), PI (Beilong Electronics Co., Ltd., Guangzhou, Guangdong, China) and PTFE (Gefang New Material Co., Ltd., Taizhou, Jiangsu, China). A single layer of PET was used as the control group, and the combination of PET and other films was used as the experimental group, resulting in a total of three groups of test samples. To create the double-layer structure, we selected a double-sided tape (3M 467MP, 3M Co., Ltd., Saint Paul, MN, USA) with a thickness of 50 μm to bond the film. When bonding is complete, the PET is first placed face up, then the surface is blackened with a marker pen (the main components were carbon black and ethanol).

First, two films of similar size are evenly bonded with double-sided adhesive. It should be noted that there should not be a large gap between the films, otherwise the stability of the structure is easily destroyed. The CPs are designed by ORIPA software (Ver. 0.35), and transferred to the laser device LaserPecker (L2, Hi-Xing Technology Co., Ltd., Shenzhen, Guangdong, China) via Bluetooth. At high temperatures, the film can produce a crease line with actual depth and width, so that there is a stiffness difference between the crease and the surface, convenient for subsequent manual folding. Finally, the end cap is assembled with the Waterbomb to achieve complete axial closure. Fabrication details can be found in the Appendix A (see Appendix A).

### 2.2. Kinematic Modeling

First, we assume that the motion of Waterbomb satisfies symmetry; the algorithmic diagram of the kinematic model can be found in the Appendix A (see Appendix A). Next, all crease intersections are classified, and the vertices are precisely positioned using the row and column numbers. Then, the kinematic equations of the Waterbomb were determined by combining three aspects: (1) the projected view of row 0 for #1 and #2; (2) the motion path of the *P* and *Q* lines; and (3) the position vector model of row 1. Finally, the complete set of kinematic equations was solved by MATLAB (R2017a). Modeling details can be found in the Appendix A (see Appendix A).

### 2.3. Finite Element Analysis (FEA)

First, a 3D model of the Waterbomb was created by SOLIDWORKS (2018 SP0.0). Then, the model was transferred to COMSOL Multiphysics^®^ 5.6. To simulate the stress of the Waterbomb under axial and radial loads, the structural mechanics module was selected and several key parameters were determined: (1) the transient solver is PRADISO; (2) t time step method is the backward difference; (3) we fixed the constraint at the bottom end cap; And (4) the axial boundary load was applied to the top end cap, and the radial boundary load was applied to the four pairs of adjacent largest triangular facets in row 0.

Finally, we evaluate the mechanical properties of #2 through three mechanical compression experiments, including quasi-static compression, cyclic compression, and creep. The experimental equipment is Universal Tensile Strength Tester (REGER GEM-10, REGER Instrument Co., Ltd., Shenzhen, Guangdong, China). In the quasi-static compression experiment, the axial and radial displacements were set at 30 mm and 12 mm, respectively, according to the actual manufacturing size of the Waterbomb (*a* = 14 mm). In the cyclic compression experiment, the number of cycles was equal to 5, and the axial and radial cycle conditions were set as periodic force (0 ≤ *F_a_
*< *F*_(2, 3)_) and periodic displacement (0 ≤ *D_r_
*≤ 12), respectively. In the creep experiment, the axial load and radial displacement remained unchanged, and the elastic deformation capacity of the film is analyzed through the change of force.

### 2.4. Pneumatic Control System

Considering the characteristics of the film and the production cost, we took the air pump of a common sphygmomanometer as the air source of the control system and designed the pneumatic actuator composed of long balloons and non-woven gauze, in which the gauze comes from the inner layer of the mask. After the inside of the Waterbomb was placed into the actuator, we secured the actuator and the end cap with a string. We connected the smart meter (YBS-80A, Xuansheng Instrument Technology Co., Ltd., Suzhou, Jiangsu, China) between the air source and the Waterbomb, and connected the RS 485 (DT-5019, DTECH Electronic Technology Co., Ltd., Guangzhou, Guangdong, China) to the computer for real-time data transmission. The pneumatic control of the Waterbomb can be found in the Appendix A.

### 2.5. Smoke Blocking Experiment

First, we processed the simulated trachea with an FDM 3D printer (Aurora Technology Co., Ltd., Shenzhen, Guangdong, China). All parts of the simulated trachea (trachea and bronchus) remain the same diameter, equal to 60 mm. Then, we chose a red smoke block (Fangcai Trading Co., Ltd., Guangzhou, Guangdong, China) composed of compressed starch and dye, which can produce a large amount of red smoke when burned. To ensure safety, the weight of each burning smoke block is not more than 5 g, and it needs to be operated in a relatively wide area. Details of the smoke-blocking experiment can be found in the Appendix A.

### 2.6. Statistical Analysis

All experiments in this study were conducted at least three times, and the data collection process was scientific and reliable. We mainly used MATLAB to process data, and the results were statistically significant. The error bar is determined by the standard error of the mean value and the error shadow band was determined by the fluctuation range of the data.

## 3. Results

### 3.1. Origami Structural Design

The CPs consist of creases and facets [54], which can be folded a finite number of times to form a three-dimensional origami structure, as shown in Figure 2a. The intersection between the creases is called the vertex, the filled area between the creases is called the facet, and the dihedral angle between the two facets with a common crease is called the folding angle. According to the direction of folding, the creases can be divided into mountain crease (M) and valley crease (V), which are shown as solid black lines and dashed lines, respectively, where the mountain crease has a folding angle *θ*_M_∈[−π, 0], and the valley crease has a folding angle *θ*_V_∈[0, π] [55]. The Waterbomb base is a typical six-crease single vertex pattern with a square outer edge, containing two colinear mountain creases and four diagonal valley creases, as shown in Figure 2b. The geometry is determined by four parameters: the side length 2*a*, the dihedral angle *θ* of the two largest triangular facets, the number of bases in the longitudinal direction *m*, and the number of bases in the circumferential direction *n*. Different proportions of *m* and *n* can form a variety of Waterbomb origami structures, also known as origami magic balls, which have significant deformability in the radial direction. The radial and axial lengths of the Waterbomb are denoted by *Lr* and *La*, respectively. In our experiments, we found that Waterbomb has excellent folding/deploying performance when 2*n* = 5*m* + 1 (*m* is odd) is satisfied, as shown in Figure 2c. As *m* increased from 3 to 11, the radial expansion and axial contraction were significantly enhanced, with an antagonistic trend in both directions. Notably, the product of the radial expansion and axial contraction ratios was almost always close to 1. Although the richer kinematic behavior is achieved at higher *m*, it makes manual folding extremely difficult. Therefore, this work only discusses the relatively simple and classic configuration with *m* = 3, *n* = 8.

To improve the Waterbomb, we proposed new CPs, as shown by (i) in Figure 2d. The pattern consists of three identical rows of eight bases closely linked, with half-columns staggered between rows to meet specific origami design principles [56]. Figure 2e shows the folded Waterbomb prototype and the backbone structure consisting of three layers: row −1, row 0, and row 1. The leftmost additional quarter column (dark gray) was glued to the rightmost column to form a closed tube, and the seal structure at the top (or bottom) was used to initially close the tube axially, as shown by (ii) in Figure 2d for the design principle. The triangles facet *P*_0_*Q*_0_*Q*_1_ and *P*_0_*Q*_0′_*Q*_1′_ overlapped each other with scissors along *P*_0_*P*_1_, and the pink area was used to reinforce the joint effect. The end cap is made up of a thin seal cap and a thick reinforced part, where the seal cap is bonded to the sealing structure, and the reinforced part was used to increase the strength of the end cap. The reinforced part was used to increase the robustness of the end cap. The end cap was geometrically constrained, limiting the ability to expand/contract simultaneously, so that the Waterbomb can only undergo positive Poisson deformation after being closed axially. Waterbombs had four configurations during the deployment process, including states #1, #2, #3, and #4, as shown in Figure 2e. State #1 is an unstable configuration that requires forces to maintain a uniform small form, and it can be quickly converted to #2 when the forces disappear. The last three are stable configurations, all with large forms. The stability of each configuration can be determined by mathematical conditions, which can be found in the Appendix A (see Appendix A). *Lr*/*a* and *La*/*a* are used to represent the dimensionless deformation standard ratios of radial and axial lengths, respectively, as shown in Figure 2f. From #1 to #4,
*Lr*_4_/*Lr*_1_ = 2.08, and *La*_4_/*La*_1_ = 0.48(3)
an interesting result can be found,
(*Lr*_4_*La*_4_)/(*Lr*_1_*La*_1_) ≈ 1(4)
it is consistent with the experimental results in Figure 2c. Formula (4) shows that the axial and radial deformations of the Waterbomb have an antagonistic relationship. Thus, although a higher radial expansion ratio is beneficial in this work, it also means that the axial needs to provide greater compensation. In addition, Formula (4) shows that #1 can be transformed into #4 by controlling only one direction, which plays a key role in actuator design.

Taking #2 as the special state of all configurations, the deformation characteristics of the Waterbomb are discussed respectively. From #1 to #2,
*Lr*_2_/*Lr*_1_ = 1.88, and *La*_2_/*La*_1_ = 0.81(5)
from #2 to #4,
*Lr*_4_/*Lr*_2_ = 1.11, and *La*_4_/*La*_2_ = 0.59(6)

On the one hand, #2 already has a large radial length; although #4 can be expanded to a larger diameter, the difference is smaller. On the other hand, both #2 and #4 are stable configurations, and the transition between them requires crossing a high energy barrier. In summary, we decided to use the deformation between #1 and #2 to achieve gas blocking.

For manufacturing, we chose thin films as the origami material [50,57,58], with the advantages of lightness, safety, flexibility, and durability, including polyethylene terephthalate (PET), polyimide (PI) and polytetrafluoroethylene (PTFE). Because of the lack of stability of single-layer films, we constructed double-layer films that are bonded with double-sided adhesive, and the creases were machined by laser machining techniques.

### 3.2. Kinematic Modeling

The deformation of the Waterbomb is extremely complex [47,59]. To systematically describe the motion mechanism, a simplified kinematic model was established in this work. Let *i* = −1, 0, 1, *j* = 0, 1, …, *n* − 1. We notice two sets of mountain creases at Waterbomb, which are located on the left and right sides of the half column and are named *P_j_* line and *Q_j_* line, as shown by (i) in Figure 3a. According to the different positions of each vertex in CPs, it can be divided into three types of six-crease intersections and two types of terminal vertices. The six-crease intersection points include *A*_*i*,*j*_, *B*_(*i*,*i*+1),*j*_ and *C*_(*i*,*i*+1),*j*_, and the terminal vertices include *P*_*i*,*j*_ and *Q*_*i*,*j*_, where *P*_*i*,*j*_ and *Q*_*i*,*j*_ are the exclusive vertices of line *P_j_* and *Q_j_* respectively, as shown by (ii) in Figure 3a. We named the plane of vertices *A*_0,*j*_ the equatorial plane, which divides the Waterbomb into the same upper and lower parts, as shown by (i) in Figure 3c. The center line that crosses vertically through the equatorial plane was defined as the rotation axis, and the symmetry axis of the *P_i_*/*Q_i_* line was defined as the reflection axis. If *θ*_Refl_ is used to represent the angle between adjacent reflection axes, *θ*_Refl_ = π/*n*, and the central angle occupied by each column in the circumferential direction was equal to 2*θ*_Refl_, as shown by (i) in Figure 3c.

If the facet and crease are regarded as rigid linkage and rotary joint respectively, then six-crease origami can be equivalent to a spherical 6R linkage with three degrees of freedom (DOF) [60,61]. Therefore, the Waterbomb formed by connecting multiple groups of six-crease vertices belongs to the multi-DOF system. To avoid using too many actuators, we make the following ideal assumptions to reduce DOF. First, creases and intersections can be equivalent to mathematical lines and points respectively. Then, the bases on the same row have the same kinematic behavior, and the bases in the same column are symmetric about the equatorial plane. Finally, the deformation process follows the zero thickness [62] and rigid origami mechanisms [43,63]. The zero thickness ignores the thickness of the facet. The rigid origami limits facet deformation, and there is no physical interference. Based on these assumptions, we only need to control half the column to achieve full motion, so the DOF of the Waterbomb can be reduced to 1.

Considering the special case when *j* = 0, 1, Figure 3b shows the motion path of *P* and *Q* lines from #1 to #2. For vertex C of the *P* line, *z_P,C_* represents the distance from vertex *C* to the axis of reflection, and *r_P,C_* represents the distance from vertex *C* to the axis of rotation. To facilitate modeling, we use the same definition for the other vertices. Note that vertex *B*/*C* and vertex *P*/*Q* determine the maximum radial and axial lengths, respectively. Thus,
*Lr* = max{2*r_P,C_*, 2*r_Q,B_*}, and *La* = max{2*z_P,P_*, 2*z_Q,Q_*}(7)
as shown by (ii) in Figure 3c, the projected view of the Waterbomb on the equatorial plane has a unique *θ*_0_ value for each state. Define *φ*_0_ as the dihedral angle of the two largest triangles adjacent facets to the base on row 0. When the Waterbomb is at #1, the folding angle *θ*_0_ has a minimum *θ*_0(min)_,
*θ*_0(min)_ = 2*θ*_Refl_, and *φ*_0_ = 0(8)
when the Waterbomb is at #2,
*θ*_0_ = 2*θ*_Refl_ + *φ*_0_, and *φ*_0_ > 0(9)
we take angle *θ*_0_ as the unique input variable of the kinematic model, *θ*_0_∈[2*θ*_Refl_, 150°]. The solution results are shown in Figure 3d, and the analysis process can be found in the Appendix A (see Appendix A). As shown by (i and ii) in Figure 3d, the variation trend of the radial and axial length of the Waterbomb. When *θ*_0_ = 2*θ*_Refl_,
(*Lr*/*a*)_min_ = 2.0, and (*La*/*a*)_max_ = 4.1(10)
when *θ*_0_ = 150°,
(*Lr*/*a*)_max_ = 5.1, and (*La*/*a*)_min_ = 2.4(11)
it can be seen that when the Waterbomb transformed from #1 to #2, the radial length expanded in a large range (2.55), while the axial length only shrunk in a small range (0.59), which is consistent with the geometric analysis above. In addition, we found that geometric conflicts existed at the junction of row 0 and row −1/1, as shown by (iii) in Figure 3d. On the one hand, *PC* = 2*CA*, and for the angle *θ_P_*_,*PC*_ the growth rate was greater than *θ_P_*_,*CA*_. According to the symmetry of Waterbomb, the vertex *B*/*C* of the *P* line cannot provide enough space in time to be compatible with *PB*/*PC*. On the other hand, the radius *r_P_*_,*P*_ of the outer circle of the end cap remained constant in length during motion. Therefore, when *θ*_0_ is large, a geometric conflict will occur at the vertex *B*/*C* of the *P* line, keeping #2 in a stable configuration. If the axial length continues to reduce, it will not only cause the *PB*/*PC* to bend, but the adjacent facets will also break the limit of rigid origami, resulting in non-rigid folding behavior. To improve the structural stability, inspired by Lee et al. [49], we designed the flexible facet at the vertex *B*/*C* on the *P* line, as shown by (i) in Figure 2d, which aims to avoid bending by reducing the length of *PB*/*PC*.

### 3.3. Mechanical Analysis

The mechanical properties of the Waterbomb can be described by eight parameters: (1) axial displacement *D*_a_, (2) axial force *F*_a_, (3) critical force *F*_a(*i*, *i*+1)_ between the stable configurations *i* and *i* + 1 in axial compression, (4) radial displacement *D*_r_, (5) radial force *F*_r_, (6) the linear regression coefficient of determination *R*^2^ of the *D*_r_-*F*_r_ curve, (7) radial stiffness *K*_r_, where *K*_r_ = *F*_r_/*D*_r_, and defining compliance as the reciprocal of *K*_r_, and (8) the rate of decrease of the axial and radial forces during the creep phase *V*_fa_, *V*_fr_, defined as the ratio of force to time.

The FEA results showed that in both axial and radial compression, the stress concentrations in the Waterbomb were at the intersections of the folds, particularly at vertex *B*/*C*, as shown in Figure 4a. This result is consistent with the geometric conflict in the kinematic analysis. To reduce stress, we added holes at all crease intersections so that the vertices became areas of actual depth after laser machining is completed, as shown by (i) in Figure 2d. In the axial direction, the stable configuration #2 can transition first to #3 and then to #4. Thus, #2 exhibited multi-stable axial quasi-static compression, as shown by (i) in Figure 4c. When *F*_a_ > *F*_a(2, 3)_, #2 can be quickly converted to #3, and the force *F*_a_ decreased significantly, which is called snap-through [51]. At this time, the Waterbomb had a negative stiffness (*K*_r_ < 0). When the thickness of origami materials was thinner, the difference in stiffness between all parts was small; #2 satisfied folding symmetry and could be converted directly to #4 without going through #3, as shown by (i) in Figure 4c (red curve). In the radial direction, #2 exhibited compliance with increased force, as shown by (ii) in Figure 4c. When the origami material was the same, the compliance decreased with the increase of the thickness of the origami material. Table 1 compares the *K*_r_ of three groups; it can be found that the double-layer structure had a greater *K*_r_ than the single layer, which contributes to the stability of the radial support.

Thin film is a kind of viscoelastic material, and its mechanical properties are affected by the number and time of deformation. To evaluate the deformability of #2 over a long period, cyclic compression and creep experiments were carried out, as shown in Figure 4d. As the number of cycles increased from 1 to 5, both axial and radial forces decreased to varying degrees, indicating that active recovery of the film lagged behind passive deformation. At the end of the five cycles, the axial force and radial displacement were kept constant, and the Waterbomb was allowed to enter the creep stage. As seen from Table 1, there was an overall positive correlation between film thickness and the rate of decrease in force (*V*_fa_ and *V*_fr_), indicating that the greater the thickness is the greater the creep rate of the sample. In addition, we found a special case in the third group of experiments, where the *V*_fr_ value of the first sample was smaller than that of the first group. The most plausible explanation for this is that manual errors led to a non-uniform distribution of the stiffness of the films, which caused a non-rigid folding behavior during compression, thus causing abnormal mechanical phenomena.

### 3.4. Pneumatic Control System

Inspired by the deformation mechanism of the McKibben artificial muscle [64], a pneumatic actuator consisting of a balloon and gauze was created, as shown in Figure 5a. Then, we wrapped an elastic band around the circumference of row 0, the purpose being to maintain the small form #1 and increase the deformation speed of #2 to #1. When *P*_0_ > 0, the actuator can only expand radially by contracting axially because of the increasing volume of the gas and the inability to extend the balloon axially, thus gaining more internal space. Figure 5b shows the pneumatic deformation process. The inflation time from #1 to #2 was approximately 2 s, while the deflation time from #2 to #1 was up to 10 s. On the one hand, unstable configuration #1 can easily change to stable configuration #2, but the transition from #2 to #1 requires overcoming a high energy barrier. On the other hand, the film is viscoelastic and has hysteresis in the deformation process, which is consistent with the result in Figure 4d. To make the control process more intelligent, the pneumatic system of real-time monitoring is constructed, as shown by (i) in Figure 5c.

The property of the actuator can be described by five parameters: (1) initial air pressure *P*_0_, (2) residual air pressure *P*, (3) air pressure variation ∆*P*, where ∆*P* = *P* − *P*_0_, (4) airtightness *P*_ratio_, where *P*_ratio_ = (*P*/*P*_0_) × 100%, and (5) the first-order fitting slope *k_x_* of the *D*_r_-∆*P* curve, where *k_x_* = ∆*P*/*D*_r_, the subscript *x* corresponding to *P*_0_ = *x* KPa, and *P*_0(max)_ ≤ 60. As seen in (ii) in Figure 5c, *P*_ratio_ reached 91.16% in 600 s, which means that the actuator has a high internal air tightness. Figure 5d–f shows the variation curves of *P*_0_ and the folding angle *θ*_0_, radial length *r*, and axial length *z* respectively. As *P*_0_ increased from 0 to 60 KPa, both *θ*_0_ and *r* had a large increase, but the axial lengths *z_P,P_* only had a small decrease. Note that the Waterbomb has an internal actuator that prevents the vertices *A_i_*_,*j*_ from being colinear with the rotation axis, and thus (2*r_P_*_,*A*_)_min_ > 0, as shown in Figure 5e. Thus, #1 is generally ellipsoidal in geometry (non-uniform diameter), and there is no exact solution to the equation *θ*_0(min)_ = 2*θ*_Refl_, as shown in Figure 5d.

To efficiently evaluate the radial mechanical properties of the Waterbomb under pneumatic control, we selected a sample with the highest radial stiffness (the first of group iii). In the inflated state, *K*_r_ is derived partly from the geometric model, and partly from the internal actuator. Within the allowable pressure range of the actuator, *K*_r_ increased as *P*_0_ increased, as shown in Figure 5g. When *P*_0_ was higher (40 ≤ *P*_0_ ≤ 60), the radial support ability was stronger, and had excellent radial mechanics. As *P*_0_ increased from 40 to 60 KPa, *K*_x_ decreased from 0.51 to 0.21, indicating the high stiffness of the Waterbomb, as shown in Figure 5h. Figure 5i shows the radial creep curve under pneumatic control. Although *V*_fr_ increased with the increase of *P*_0_, *V*_fr_ could be kept within a small range, making the Waterbomb made of thin film safe and controllable. Finally, the mechanical data obtained from the abovementioned analysis are shown in Table 2.

### 3.5. Simulated Trachea Experiment

Based on the above-shown analysis, we built a prototype of Oribron, as shown by (i) in Figure 6a. When the gas enters from the near end of RBs, #1 becomes #2, as shown by (ii) in Figure 6a. By adjusting the pneumatic control valve, the Waterbomb can easily obtain any configuration between #1 and #2. We processed the simulated trachea with 3D printing technology, the geometric shape was Y-shaped, including the trachea and bronchus, as shown in Figure 6b. Then, we put the Waterbomb into the trachea, and compared the changes in the main/top view to evaluate the radial support capability. When the Waterbomb was at #1, there was a large gap between row 0 and the trachea, as shown by (ii) in Figure 6c. Therefore, #1 enters and exits the trachea smoothly, as shown in Figure 6c. When the Waterbomb was at #2, row 0 was infinitely close to the trachea, so #2 could maintain close contact with the trachea without force, as shown by (ii) in Figure 6d.

Because of the limitation of manufacturing technology, this work did not achieve the desired effect as shown in Figure 1a (Waterbomb connected to the body), meaning that the gas could not be smoothly transported to the distal. On the other hand, Oribron has more gaps, making it difficult for the sensors to accurately monitor pressure changes. Therefore, we designed a smoke-blocking experiment to evaluate the radial support capability of Oribron, as shown in Figure 6e.

First, we divided the experiment into three cases, respectively, in the tracheal segment of the simulated trachea: no obstruction, placement #1, and placement #2. Notice that the other side of the bronchus was closed so that the smoke can only move toward the trachea. Then, the burning red smoke block was placed into the collection device, and the device outlet was connected to the open bronchial segment. Finally, the smoke passed into the simulated trachea. The experimental results showed that a large amount of smoke escaped from the trachea segment in the first and second cases, as shown by (i and ii) in Figure 6e. In the third case, only a small amount of smoke appeared in the tracheal segment, as shown by (iii) in Figure 6e. Therefore, when #1 was converted to #2, the radial support capacity was significantly improved, and the smoke escape was effectively blocked.

## 4. Discussions

Oxygen saturation (SpO_2_) is an important physiological parameter for the transport of oxygen by blood [65], defined as the ratio of oxyhemoglobin to all oxygen-carrying hemoglobin. When oxygen saturation is below 90%, it is considered abnormal [66]. In recent years, the world has been severely affected by Coronavirus Disease 2019 (COVID-19) [67,68], and some immunocompromised people are prone to recessive hypoxia after infection with COVID-19 [69]. Although there is no obvious discomfort, the SpO_2_ is well below normal and they are in critical condition. RBs are an important tool for the clinical treatment of hypoxia, which can provide a large amount of oxygen to reduce the damage caused by hypoxia. Compared with the traditional RBs, Oribron can achieve rapid and large multiple radial deformation with oxygen supply. When *P*_0_ is higher, the Waterbomb gradually maintains a large form of high stiffness, and the gap between the backbone and trachea decreases continuously. At the same time, the Waterbomb effectively reduces the loss of oxygen, thus promoting the synthesis of oxygenated hemoglobin. As a result, Oribron has great potential to improve SpO_2_.

Nevertheless, there were some limitations to this work, as follows:(1)Miniaturization. In theory, the origami method has scale and material independence [70,71,72]. Therefore, any material can be used to make deformable structures of any size if it meets the requirements of processing technology. However, there are still great challenges in practice. On the one hand, the seal structure of a small area needs strong adhesive double-sided tape or glue; otherwise, it is easy to destroy the structural stability, and the symmetry of motion. On the other hand, the small-size actuators need to meet high machining accuracy to accommodate the narrow internal space of #1;(2)Intelligence. This work used a soft tube to transfer air from a fixed air source to achieve control, lacking an intelligent self-perception and communication transmission ability. Therefore, the integration of lightweight, multifunctional intelligent systems on thin films could be considered in the future, thereby extending the range of applications.

Finally, we expect Oribron to be difficult to compete with commercially available medical devices at present, but the origami method will continue to promote the development and adoption of related fields.

## 5. Conclusions

(1)Multifunctional deformable RBs were proposed. Compared with the traditional RBs, they can rapidly improve the utilization rate of oxygen in the oxygen supply and have great application potential in the treatment of hypoxia;(2)Improved Waterbomb CPs were proposed. The folded three-dimensional structure had excellent radial support capability and effectively blocked the escape of smoke in the simulated trachea;(3)A complete origami kinematic model was established. The model accurately describes the motion path of Waterbomb, and the causes of geometrical conflicts which affect the stability of the structure were found;(4)A pneumatic artificial muscle actuator (DOF = 1) was designed. The actuator was highly airtight and had a lower cost to manufacture, allowing flexibility to control the deformation process of the Waterbomb.

## Figures and Tables

**Figure 1 micromachines-14-00822-f001:**
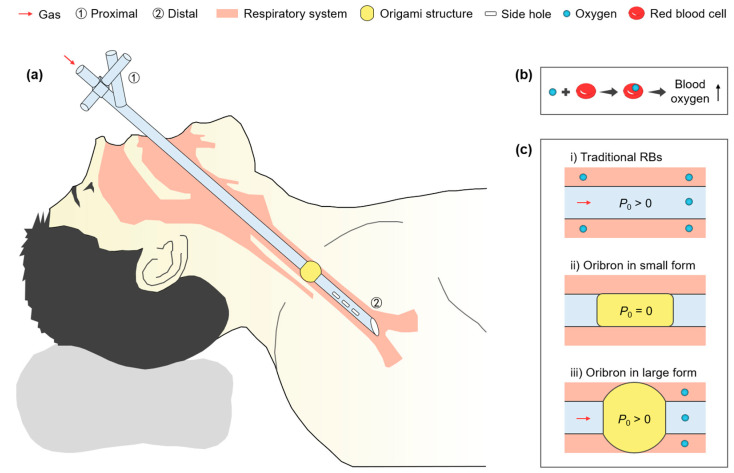
The design principle of Oribron. (**a**) Application. The blue area represents traditional RBs and together with the yellow area form Oribron. (**b**) The principle of increasing blood oxygen. (**c**) Oxygen supply effect of the traditional RBs and Oribron.

**Figure 2 micromachines-14-00822-f002:**
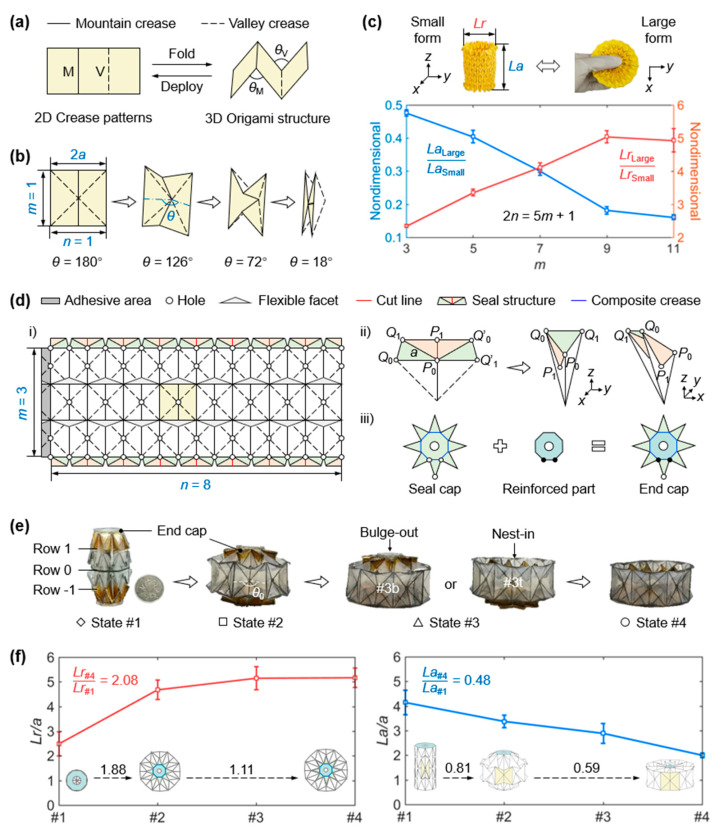
Origami design and analysis. (**a**) Origami mechanism. (**b**) The folding sequence of the Waterbomb base. (**c**) Five groups of Waterbomb satisfy the equation 2*n* = 5*m* + 1. (**d**) Waterbomb (*m* = 3, *n* = 8). (i) CPs; (ii) seal structure; (iii) End cap. (**e**) Deploying sequence of the Waterbomb. Materials: backbone (PET + PTFE), end cap (PET + PI). (**f**) Standard ratios of radial and axial length.

**Figure 3 micromachines-14-00822-f003:**
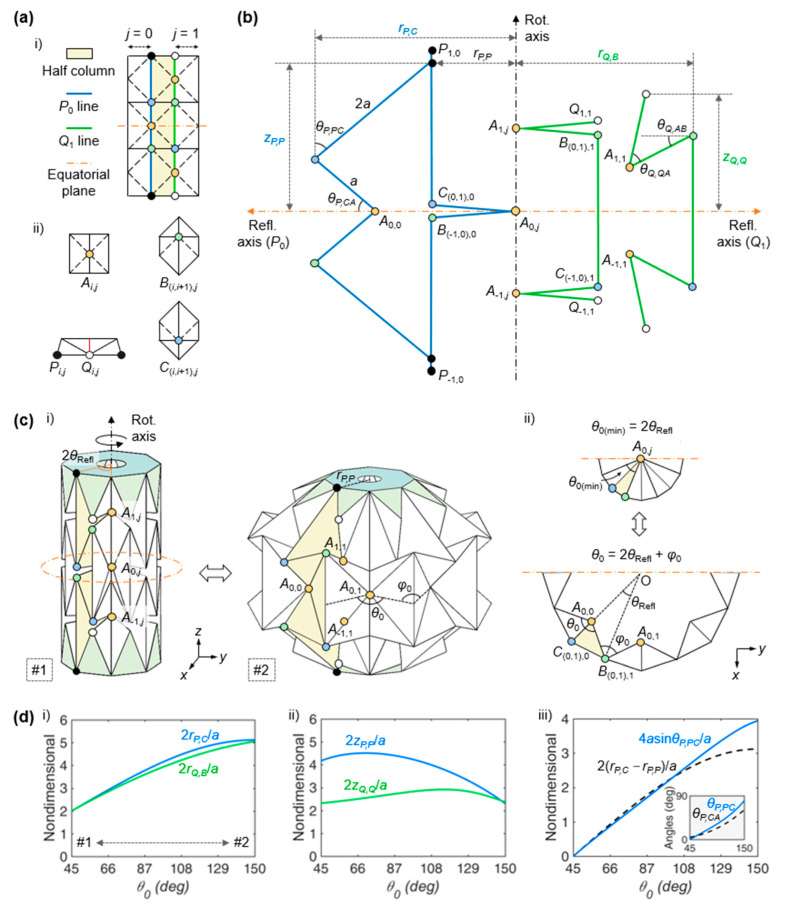
Kinematic model of the Waterbomb. (**a**) Vertex classification. (i) half-column of CPs; (ii) five types of vertices, represented by solid circles in yellow, green, blue, black, and white. (**b**) The motion paths of the *P*_0_ and *Q*_1_ lines. (**c**) #1 and #2. (i) main view; (ii) half-column of the projected view on the equatorial plane. (**d**) Model solution. (i) radial length; (ii) axial length; (iii) geometric conflict.

**Figure 4 micromachines-14-00822-f004:**
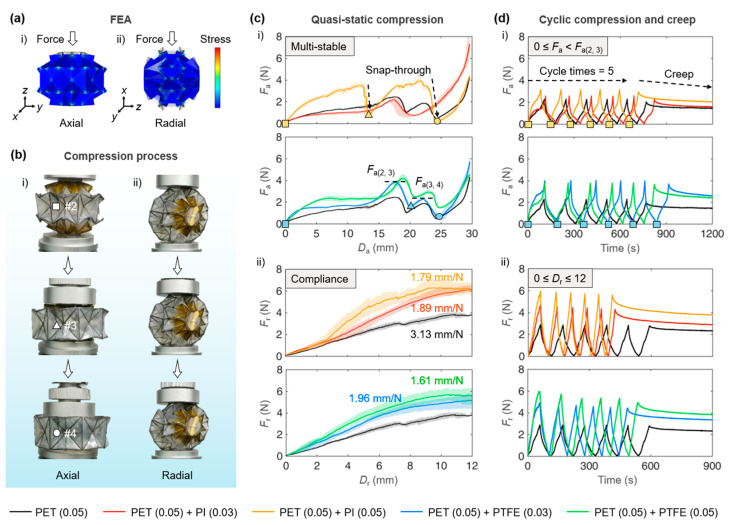
Mechanical compression properties of the Waterbomb (#2). (**a**) FEA results. (i) axial; (ii) radial. (**b**) Compression process. (i) axial; (ii) radial. (**c**) Quasi-static compression, and the shaded area near the curve representing the error band. (i) axial; (ii) radial. (**d**) Cyclic compression and creep experiments. (i) axial; (ii) radial.

**Figure 5 micromachines-14-00822-f005:**
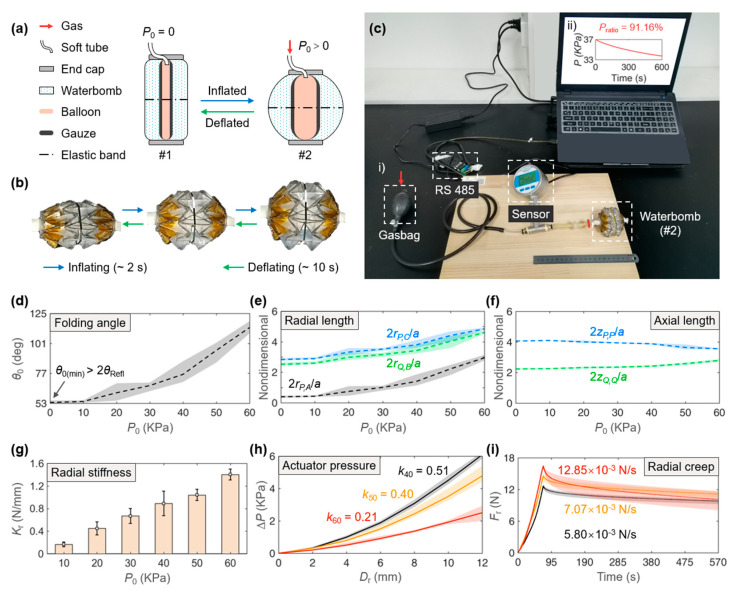
Pneumatic control system. (**a**) Design principle of the actuator. (**b**) Pneumatic deformation process. (**c**) Pneumatic devices. (i) components of the device; (ii) airtightness of the actuator within 600 s. (**d**) Folding angle *θ*_0_. (**e**) Radial length *r*. (**f**) Axial length *z*. (**g**) Radial stiffness *K*_r_. When *P*_0_ is equal to 40, 50, and 60 KPa, see (**h**) for actuator stability and (**i**) for the radial creep experiment.

**Figure 6 micromachines-14-00822-f006:**
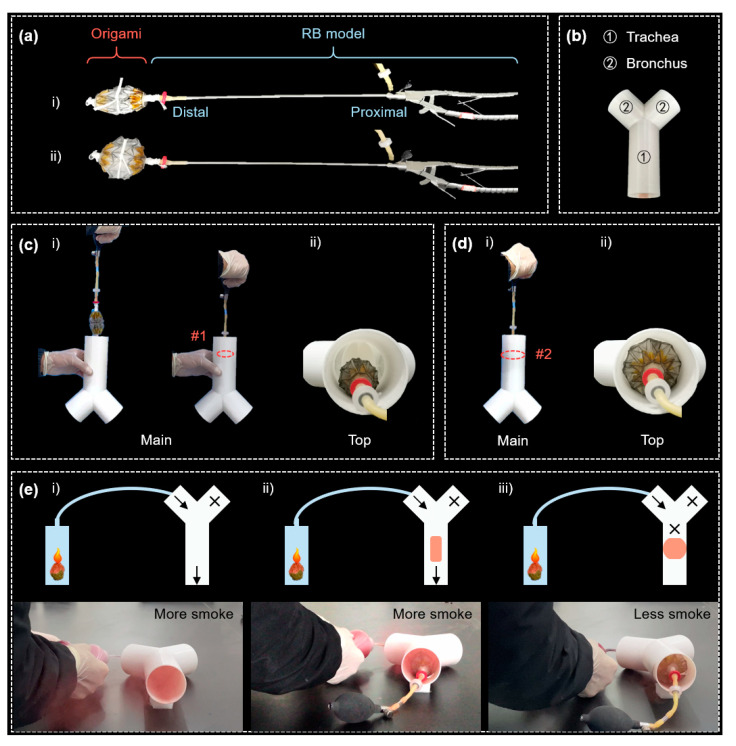
Radial support capability of the Waterbomb. (**a**) Oribron device. (i) when the Waterbomb is at #1, and (ii) when the Waterbomb is at #2. (**b**) Simulated trachea. (**c**) Radial support tests inside the trachea for #1: (i) main view; (ii) top view. (**d**) Radial support tests inside the trachea for #2: (i) main view; (ii) top view. (**e**) Smoke block experiment. From left to right inside the trachea: (i) no obstruction is placed; (ii) place #1; and (iii) place #2.

**Table 1 micromachines-14-00822-t001:** Mechanical properties of the Waterbomb.

Group	Thickness of Materials(mm)	Mass (g)	*R* ^2^	*K*_r_ (N/mm)	*V*_fr_ (10^−3^ N/s)	*V*_fa_ (10^−3^ N/s)
i	PET (0.05)	3.96	0.98	0.32	1.40	1.47
ii	PET (0.05) + PI (0.03)	6.18	0.99	0.53	2.98	1.57
	PET (0.05) + PI (0.05)	7.11	0.93	0.56	3.70	1.95
iii	PET (0.05) + PTFE (0.03)	8.47	0.90	0.51	1.38	3.17
	PET (0.05) + PTFE (0.05)	9.36	0.97	0.62	3.15	3.30

**Table 2 micromachines-14-00822-t002:** Radial mechanics of the Waterbomb under pneumatic control.

*P*_0_ (KPa)	*K*_r_ (N/mm)	∆*P*/*D*_r_ (KPa/mm)	*V*_fr_ (10^−3^ N/s)
40	0.89	0.51	5.80
50	1.04	0.40	7.07
60	1.40	0.21	12.85

## Data Availability

All data required to support the conclusions of this paper are included in the text and Appendix A. Y.D. can be contacted for further information.

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
