# Peer review of "Oribron: An Origami-Inspired Deformable Rigid Bronchoscope for Radial Support"

_micromachines, 2023, doi:10.3390/mi14040822_

Round 1
Reviewer 1 Report
This study proposed a deformable bronchoscope (named Oribron) by added Waterbomb origami structure. Oribron is promising to improve the utilization rate of oxygen during the use of bronchoscope. The reviewer has the following comments:
1. In the abstract, it is not well described how does Oribron improve the utilization rate of oxygen.
2. Are there other ways or devices that can improve the utilization rate of oxygen during the use of bronchoscope? Please compare with the existing research and give an analysis.
3. This paper mentions that the design of Water-bomb has good performance when 2n=5m+1 is satisfied, but the specific reason, experimental process and experimental data are not found.
4. Please check and revise the grammar, typesetting and punctuation errors.
Reviewer 2 Report
The study presented in the manuscript is of medical and technical interest and can be recommended for publication after its presentation has been improved.
1. It is recommended to refer to formulas in the text as follows: From (1) to (2) and not From #1 to #2.
2. It is recommended to use more convenient variable notation, such as Lr4 rather than Lr#4.
3. It is recommended to provide a detailed diagram of the Waterbomb kinematic analysis algorithm.
4. Page 6 discusses the issue of stable and unstable Waterbomb configurations. Which configuration should be considered as stable and which as unstable. What mathematical conditions make it possible to judge the stability of the Waterbomb configuration?
5. On the page 6 it is stated that when the Waterbomb is deployed, 4 geometric device configurations are possible, one of which is unstable and three are stable. What kind of configuration will be obtained during expansion and how to confirm this?
6. What exactly is the interest in formula (4)?
Round 2
Reviewer 1 Report
In my opinion, this paper will be accepted now.